# The Cyst Epithelium in Polycystic Kidney Disease Patients Displays Normal Apical-Basolateral Cell Polarity

**DOI:** 10.3390/ijms25031904

**Published:** 2024-02-05

**Authors:** Samuel Loft Sandegaard, Andreas Riishede, Henrik Birn, Helle Hasager Damkier, Jeppe Praetorius

**Affiliations:** 1Department of Biomedicine, Health Faculty, Aarhus University, DK-8000 Aarhus C, Denmark; sls@biomed.au.dk (S.L.S.); and@clin.au.dk (A.R.); hb@clin.au.dk (H.B.); hd@biomed.au.dk (H.H.D.); 2Department of Clinical Medicine, Health Faculty, Aarhus University, DK-8200 Aarhus N, Denmark

**Keywords:** polycystic kidney disease, cell polarity, Crumbs complex, Par complex, Scribble complex, cadherin

## Abstract

The main characteristic of polycystic kidney disease is the development of multiple fluid-filled renal cysts. The discovery of mislocalized sodium-potassium pump (Na,K-ATPase) in the apical membrane of cyst-lining epithelia alluded to reversal of polarity as a possible explanation for the fluid secretion. The topic of apical Na,K-ATPase in cysts remains controversial. We investigated the localization of the Na,K-ATPase and assessed the apical-basolateral polarization of cyst-lining epithelia by means of immunohistochemistry in kidney tissue from six polycystic kidney disease patients undergoing nephrectomy. The Na,K-ATPase α1 subunit was conventionally situated in the basolateral membrane of all immunoreactive cysts. Proteins of the Crumbs and partitioning defective (Par) complexes were localized to the apical membrane domain in cyst epithelial cells. The apical targeting protein Syntaxin-3 also immunolocalized to the apical domain of cyst-lining epithelial cells. Proteins of the basolateral Scribble complex immunolocalized to the basolateral domain of cysts. Thus, no deviations from the typical epithelial distribution of basic cell polarity proteins were observed in the cysts from the six patients. Furthermore, we confirmed that cysts can originate from virtually any tubular segment with preserved polarity. In conclusion, we find no evidence of a reversal in apical-basolateral polarity in cyst-lining epithelia in polycystic kidney disease.

## 1. Introduction

Autosomal polycystic kidney disease (PKD) is a potentially lethal monogenic disorder characterized mainly by the development of fluid-filled cysts in the kidneys of patients and various extra-renal manifestations [1]. The disease is associated with mutations in the genes *PKD1* and *PKD2*, which encode polycystin 1 (PC1) and polycystin 2 (PC2), respectively. These proteins are located in or near the primary cilium of tubule cells, and their dysfunction leads to abnormal cell proliferation and fluid secretion [2]. 

In 1991, Wilson et al. found Na,K-ATPase to be mislocalized in the apical membrane of cyst-lining epithelia [3]. This provided a possible explanation for the phenomenon of fluid secretion: Perhaps the epithelial cells were secreting rather than reabsorbing fluid because of a reversal of their apical-basal polarity. Cell polarity was further implicated in the pathogenesis of polycystic kidney disease by the discovery of aberrant planar cell polarity in animal models of PKD [4,5]. 

The establishment and maintenance of epithelial apical-basolateral cell polarity relies on three things: intrinsic sorting of membrane proteins to different segments of the membrane, protein complexes situated at the respective membrane domains, and extrinsic stimuli that inform the cells of their orientation in the three-dimensional space [6].

Intrinsic sorting is based on motifs inherent to the respective proteins as well as membrane properties. For instance, tyrosine-based and dileucine motifs in the cytosol-facing domains of membrane proteins are associated with basolateral sorting, while GPI anchors and glycosylation are associated with sorting to the apical membrane [7]. The docking of vesicles to membranes is dependent on interaction between so called Soluble *N*-ethylmaleimide-Sensitive Factor Attachment Proteins (SNAP) receptors (SNAREs), v-SNARE and t-SNARE complexes. The apical and basolateral membranes differ in that syntaxin-3 is present in the t-SNARE complexes of the apical membrane, while syntaxin-4 is expressed in the basolateral membrane [8,9]. Furthermore, lipid composition may have implications for vesicle docking and protein trafficking, as phosphatidylinositol 3,4,5-trisphosphate (PIP_3_) is present in the basolateral membrane, and phosphatidylinositol 4,5-bisphosphate (PIP_2_) is found mainly in the apical membrane [10,11].

The polarity factors defining the apical domain are the portioning defective (Par) protein complex, consisting of atypical Protein kinase C (aPKC), Partitioning defective protein 6 (Par-6), Cell division control protein (CDC-42), and the Crumbs complex, comprised of Protein Associated With LIN7 1 (Pals-1) and PALS1-associated tight junction protein (PATJ) [12,13]. The aPKC forms a complex with Par-6 which allows aPKC to exert its phosphorylating properties [14]. This activity is amplified by the binding of CDC-42 and the Crumbs complex [15]. The resultant phosphorylation is the basis for the exclusion of proteins from the apical domain [16,17]. The basolateral domain is defined by the polarity factors Scribble, discs large (Dlg), and lethal giant larvae (Lgl), collectively referred to as the Scribble complex [12]. The tight junction constitutes a molecular fence separating the apical from the basolateral domain, thereby hindering the lateral diffusion of membrane proteins belonging to the apical into the basolateral domain [18]. 

Additionally, in order to polarize correctly, cells require extrinsic signals such as cell–cell interaction via E-cadherin [19] and extracellular matrix-cell interaction with laminin [20]. 

Thus, several molecules involved in cell polarity offer the opportunity to investigate the polarization of epithelia by means of immunohistochemistry. This is of special interest in the context of PKD, since a better understanding of pathogenesis, cystogenesis, and fluid secretion might offer new ways to treat disease progression. In this study, we investigated whether the apical-basolateral polarity of cyst-lining epithelia had been disturbed in kidney tissue from six PKD patients. 

## 2. Results

The regular epithelial single cuboidal cell layer lining of most cysts was replaced by apparently dedifferentiated and flattened epithelium in a subset of cysts in the human PKD sections (hPKD) [21]. These cysts were void of almost any immunohistochemical staining. In some cases, staining was observed. However, the thinness of the cells made it impossible to distinguish the subcellular localization of the signal. Thus, assessment of the apical-basolateral polarity of the thin-walled cysts was hindered and, in the following, only cysts lined with taller epithelium are considered. 

### 2.1. The Na,K-ATPase α1 Subunit and E-Cadherin Localize to the Basolateral Membranes of Renal Cyst Epithelia

Previous studies indicated that the Na,K-ATPase is expressed in luminal domains of renal cysts from human kidneys, thus indicating a reversed cell polarity [3,22]. As exemplified in Figure 1A, immunolabelling for the Na,K-ATPase α1 subunit revealed the expected basolateral domain staining in proximal tubules and distal tubules. All stained cyst epithelia also displayed basolateral expression Na,K-ATPase (Figure 1B,C). Apical immunolabelling of cysts was never observed in any of the 6 biopsies. E-cadherin is another example of a typical epithelial membrane protein from the basolateral domain [23]. Figure 1D shows that the cell–cell adhesion molecule E-cadherin was also immunolocalized to the basolateral membrane domains of renal tubules, with the most prominent signal being found in the collecting ducts. In E-cadherin-positive renal cysts (the majority of cysts), the protein was confined to the basolateral plasma membrane domain (Figure 1E,F). Thus, we did not find evidence of reversed polarization of the Na,K-ATPase and E-cadherin in human renal cysts from 6 biopsies. We note that despite a marked unspecific reaction, there was no specific staining signal for E-cadherin in the proximal tubule.

### 2.2. SNARE-Protein Syntaxin-3, Crumbs and Par Complexes Are Confined to the Apical Membranes of Renal Cyst Epithelia

Immunoreactivity against the apical SNARE protein Syntaxin-3 [24] was observed in the apical plasma membrane domain of mainly the proximal renal tubules, as shown in Figure 2A. Some cysts were also positive for apical Syntaxin-3 (Figure 2B,C). 

The Crumbs and Par complexes constitute the defining apical membrane domain protein complexes [6,11]. Figure 2D illustrates apical membrane immunoreactivity for Crumbs-3 in proximal and distal renal tubules. Apical Crumbs-3 staining was observed in all immunoreactive cysts (Figure 2E,F). Apical immunostaining of the other Crumbs complex proteins Pals-1 (majority of cysts) and PATJ was also observed in renal tubules as well as in all immunoreactive cysts (Figure 3A–C and Figure 3D–F, respectively).

The Par complex is also an apical and tight junctional protein complex typically consisting of Par-3, Par-6, and CDC-42 [13,25,26]. Only the Par-6 and CDC-42 antibodies revealed immunostaining in human kidney biopsies. Figure 4A illustrates apical membrane domain Par-6 immunoreactivity in proximal tubules and, less convincingly, in distal renal tubules and collecting ducts. The Par-6 positive cyst epithelium also exhibited exclusive apical membrane labelling (Figure 4B,C). Thin-walled cysts occasionally displayed basal membrane Par-6 immunostaining (not shown). CDC-42 immunoreactivity was observed in renal tubules corresponding to the apical plasma membrane domain (Figure 4D). A similar labelling pattern was observed in all immunoreactive renal cysts (Figure 4E,F). We note that the proximal tubule did not exhibit staining for CDC-42, while the thick ascending limb (TAL) did not show immunoreactivity for Par-6 or CDC-42. Neither did we detect any CDC-42 signal from the cortical collecting ducts. Taken together, all detected apical membrane determinant proteins were expressed in the apical membrane of renal cyst epithelial cells.

### 2.3. Scribble Complex Components Are Expressed in the Basolateral Domain of Renal Cysts

Scribble, Lgl-1, and Dlg-1 belong to the basolateral domain Scribble complex [27]. Scribble immunoreactivity was observed in the basolateral membrane domain of distal renal tubules and collecting ducts (Figure 5A). Figure 5B,C shows similar basolateral domain labelling for Scribble in cysts. Immunolabelling for Lgl-1 also yielded a basolateral domain staining pattern in distal renal tubules and collecting ducts (Figure 5D), which was likewise observed in the immunoreactive cysts (Figure 5E,F). Tubular Dlg-1 immunolabelling was not obtained. Accordingly, we did not find evidence to support the reversal of epithelial cell polarity in human renal cysts. We note that proximal tubules and thick ascending limbs were devoid of the staining signal of Scribble and Lgl-1.

### 2.4. Renal Cysts Origin from More Tubular Segments

Renal cysts can arise from virtually all parts of the renal tubular system [28,29]. To assess the origin of cysts in the 6 biopsies, antibodies were applied to identify cysts with the proximal tubule marker: the water channel aquaporin-1 (AQP1),as only renal cortex is present, the thick ascending limb (TAL) marker: sodium-potassium-chloride exchanger 2 (NKCC2), the distal tubule markers: the sodium–chloride cotransporter (NCC), and parvalbumin, as well as the collecting duct marker: the water channel aquaporin-2 (AQP2). As illustrated in Figure 6A, AQP1 is expressed in proximal tubules with staining mainly in the basolateral domain. Robust labelling was also observed in capillary endothelia. Cysts were rarely identified with AQP1 immunoreactivity, but positive cysts displayed both apical and basolateral membrane labelling (Figure 6B,C). NKCC2 immunoreactivity was observed in the apical membrane domain of cortical tubules (Figure 6D) as well as in NKCC2-immunoreactive cysts (Figure 6E,F). Tubular NCC immunoreactivity was found exclusively at the apical domain of immunoreactive distal renal tubules (not shown). NCC immunoreactivity was observed in rare cysts apparently with staining of the underlying basement membrane and was therefore considered unspecific staining (not shown). Parvalbumin was also expressed in distal renal tubules in the expected cytosolic pattern (Figure 7A). Rare parvalbumin-positive cysts also displayed cytosolic staining (Figure 7B,C). AQP2 immunoreactivity was observed in collecting ducts with the main reaction corresponding to the apical membrane domain (Figure 7D). Renal cysts also displayed apical membrane domain labelling for AQP2 (Figure 7E,F). Thus, except for flat cyst epithelia, renal cysts expressed markers for at least one of the markers of proximal tubules, thick ascending limbs, distal convoluted tubules, or collecting ducts. Double fluorescence immunolabelling did not reveal cysts that were positive for more than one tubule marker (not shown).

### 2.5. Preserved Cell Polarity in Cysts Derived from Different Segments

To determine whether cell polarity was conserved in cysts of all origins, double immunofluorescent labelling was employed.

Figure 8A shows the normal apical localization of Crumbs complex component Pals-1 in proximal tubules stained for AQP1. The same staining pattern is evident in the cyst in Figure 8B. Lgl-1 of the Scribble complex is shown in Figure 8C to be basolaterally localized in normal human proximal tubule. This pattern is reproduced in cysts positive for AQP1 (Figure 8D). 

Due to difficulties in obtaining satisfactory double-labelling, we employed another marker for the thick ascending limb, Claudin-16. In normal human thick ascending limb, the Crumbs complex component Pals-1 is apically immunolocalized (Figure 9A). The same localization is apparent in the Claudin-16-positive cysts (Figure 9B). Lgl-1 localizes to the basolateral domain in both normal human tubules and in the TAL-derived cysts (Figure 9C,D). We note that there is no signal for Lgl-1 in the normal TAL (Figure 9C), while a clear basolateral signal is evident in the NKCC2-positive cyst (Figure 9D).

In the normal human distal tubule, a mosaic pattern of parvalbumin is observed alongside apical localization of Crumbs complex component Pals-1 (Figure 10A). A similar pattern is found in cysts assumed to originate from the distal tubule (Figure 10B). In a normal human distal tubule, staining for E-cadherin reveals a basolateral pattern (Figure 10C). This same pattern is seen in parvalbumin-positive cysts (Figure 10D). 

With AQP2 acting as a marker for the collecting duct, the expected apical localization of Pals-1 is confirmed in this segment in normal human tissue (Figure 11A) as well as in AQP2-positive cysts (Figure 11B). Figure 11C shows the basolateral localization of E-cadherin in a normal human collecting duct. This pattern is repeated in cysts derived from this segment (Figure 11D). 

## 3. Discussion

It is widely accepted that cysts develop from virtually any segment of the nephron as well as the collecting duct. However, they most often develop from the distal parts, which is consistent with our finding that AQP1-positive cysts were rare [29]. According to the current paradigm, most cysts will detach from their tubule of origin and fluid will accumulate in the cyst lumen [21]. This accumulation is attributed to the secretion of ions from the cyst epithelium with concurrent water transport to the lumen. Some researchers have found an apical mislocalization of the Na,K-ATPase in cyst-lining epithelia, which was suggested to be responsible for Na^+^ secretion with resultant water accumulation in the cysts [3,22,30]. Other groups report no such apical localization of the Na,K-ATPase in either human PKD kidneys or in animal models [31,32,33,34]. The results of the present study are in accordance with the latter in that we found no apical staining for the Na,K-ATPase α1 subunit in normal tubules or cyst epithelia. Thus, our findings suggest a different secretion mechanism. It has been proposed that the key secreted ion is chloride instead of sodium—the argument being that dysfunctional polycystin 2 (PC2) leads to lower [Ca^2+^]_i_. This, in turn, presumably leads to a rise in intracellular cAMP through lacking inhibition of calcium-inhibited adenylate cyclase isoforms 5 and 6 (AC-5/6) [35,36]. The cAMP then stimulates the activity of the cystic fibrosis transmembrane conductance regulator (CFTR) through which chloride will be transported to the cyst lumen followed by Na^+^ and water through a paracellular route [37,38]. Our findings do not exclude this secretion of Cl^−^ but argue against the proposed active secretion of Na^+^. This cAMP-dependent secretion is, however, confined to the segments distal to the loop of Henle. Thus, the accumulation of fluid in cysts derived from proximal tubule or TAL remains elusive.

To determine whether the change in transport direction of the cyst-lining epithelium results from a simple reversal of the basic cell polarity, we investigated the expression of the crucial polarity proteins in these cells. The zonula adherens molecule E-cadherin was shown to be correctly situated on the basolateral side of the cyst-lining epithelium. Apical markers such as the SNARE-protein Syntaxin-3 as well as the Crumbs and Par complexes were also found to be correctly localized in cyst epithelia. The basolateral Scribble complex was also found in the expected location in cysts. Taken together, we found no evidence of any polarity switch as a part of cystogenesis in hPKD. 

The study is limited to analysis of tissue from only six patients. This smaller sample size naturally gives rise to the question of whether the samples are representative of the entire PKD patient group with regard to clinical manifestations and mutation status. Due to the anonymity of the patients and the terms of our license, we are unable to retrieve any such information. Within this limited group of patients, we are, however, able to confidently state that there are no inconsistencies with regard to the apical-basolateral polarity in cyst epithelia. 

It is important to keep in mind that the investigated tissue is but a small percentage of the entire patient kidney. Due to the nature of the tissue collection, we are only able to assess cysts of a certain size. In our sections, intact cysts with a diameter of up to 1 cm are present. In some sections, portions of seemingly larger cysts are detected. We cannot, however, know their true size. Whether or not cysts are detached from their tubular origin is also not possible to determine with our immunohistochemical analysis. Our methods do not allow us to provide the exact number of unique cysts assessed, since some sections will carry portions of cysts also appearing in other sections. Additionally, we would be at risk of underestimating the number of cysts, due to our conservatism in designating cysts as such. 

In accordance with the literature, we found cysts to originate from virtually every tubule segment [28,29]. It is worth noting, however, that we could not achieve any satisfactory staining for the apical distal tubule transporter NCC in cysts. On rare occasions, staining was observed beneath the basement membrane and was therefore regarded as unspecific staining. Instead, we used parvalbumin as a marker for the distal tubule, which was also present in cysts. No cysts were positive for multiple segment markers, indicating a singular segmental origin. Cysts originating from proximal tubule, TAL, distal tubule, and collecting ducts were all found to have preserved planar polarity. Statistics on the origin of cysts were intentionally omitted. Since the analyzed tissue is a smaller fraction of the entire kidney, the resultant statistical analysis would be applicable only to the surgical biopsies and not the whole kidney. 

Not all tubule segments express all polarity proteins. The proximal tubule did not exhibit staining for the basolateral markers E-cadherin, Scribble, or Lgl-1 by means of immunoperoxidase. The proximal tubule also did not stain for apical CDC-42, while Syntaxin-3 was only observed in this segment (Table 1). 

The absence of E-cadherin from the proximal tubule is not surprising, since N-cadherin is expressed in the proximal tubule, while E-cadherin is present in the distal parts of tubule system [39]. This is without consequence to the interpretation of our data, since the cysts arising from the proximal tubule are shown to have conventional polarity with regards to Pals-1 and Lgl-1. Strikingly, Lgl-1 was observable by means of immunofluorescence.

The presence of Syntaxin-3 only in the proximal tubule is surprising seeing as syntaxin-3 has previously been described to only be present in the basolateral membrane of intercalated cells of the collecting duct [40]. The same group later found Syntaxin-3 to be expressed in the basolateral membrane in all segments except the proximal tubule [41]. Since we found Syntaxin-3 to be apically localized in both normal human tubules and cyst epithelium, this discrepancy does not alter the interpretation of our results.

Lgl-1 was not detected in the thick ascending limb, but, surprisingly, was present in NKCC2-positive cysts. We interpret this as cysts originating from TAL but losing their phenotype. Importantly, Lgl-1 was basolaterally expressed as expected. Thus, this finding aids us in determining the preserved polarity of the NKCC2-positive cyst.

As mentioned in the results section, we were only able to achieve detectable immunohistochemical staining from cysts lined with a relatively tall epithelium. The thin-walled cysts exhibited no immunoreactivity, which hindered our assessment of the polarity and tubular origin of these cysts. It is possible that this tapering of the epithelium represents a gradual de-differentiation from the original tubular phenotype [21]. As such, we are unable to comment on whether this proposed de-differentiation also results in aberrant polarization or apical expression of Na, K-ATPase. We would argue, however, that the thinning of the cyst-lining epithelium more likely signifies a lower functional state with regards to reabsorption than it represents a switch of cell polarity.

Since it has been shown that complete loss of the primary cilium leads to cyst development [42], it might be valuable to investigate whether there is a connection between cyst origin and cilium loss. In the present study, we find no evidence of a polarity reversal in the cyst epithelium, and as such, the fluid accumulation in cysts cannot be explained by this. Therefore, it would be interesting to see if the cysts all express CFTR in the apical membrane domain and NKCC1 in the basolateral membrane with preserved polarity, as suggested by the hypothesis of cyst fluid secretion [37,43]. In order to identify targets for managing cyst formation, more studies are warranted to elucidate how cyst morphology correlates to its secretory state and to the expression of plasma membrane ion transport proteins.

In conclusion, we confirm the principle that renal cysts can arise from the entire tubular system—but only express membrane transport markers of a single segment. All cysts with cuboidal epithelial cells are polarized by the conventional lateral membrane domain cadherin expression, apical Par-6, syntaxin-3, and Crumbs complexes, as well as basolateral Scribble complex. Furthermore, the Na,K-ATPase expression was confined to the basolateral membrane of all the cuboidal cyst-lining epithelia. Thus, this study cannot support a model of secretory cyst epithelium relying on reversed cell polarity and apical mislocalization of the Na,K-ATPase. Thus, other avenues than cell polarity must be taken in the hope of future initiatives to prevent or delay cyst formation in PKD.

## 4. Materials and Methods

### 4.1. Preparation of Patient Material

Two renal surgical biopsies of approximately 0.5 × 1 × 1 cm from 6 anonymous polycystic kidney disease patients (3 males and 3 females aged 39–65 years) were taken from one kidney and immersion fixed with 4% paraformaldehyde in phosphate buffered solution (PBS) overnight. Biopsies were dehydrated in graded ethanol (70%, 96%, and 99%) for 2 h each and were left overnight in xylene. The tissue was embedded in paraffin wax, cut into 2 μm thick sections on a rotary microtome (Leica Microsystems, Wetzlar, Germany), and placed on Super Frost slides. 

### 4.2. Immunohistochemistry

Sections were dewaxed in xylene and rehydrated in graded ethanol. Endogenous peroxidase was blocked after 96% ethanol in 33% H_2_O_2_ in methanol. To retrieve antigens, sections were boiled in a microwave oven in buffer with 10 mM Tris and 0.5 mM ethylene glycol-bis(β-aminoethyl ether)-N,N,N′,N′-tetraacetic acid (EGTA), pH 9. Aldehydes were quenched in 50 mM NH_4_Cl in PBS, and the sections were blocked in 0.1% bovine serum albumin (BSA), 0.2% gelatine, and 0.05% saponin in PBS. Finally, sections were incubated overnight at 4 °C with primary antibody (Table 2) in PBS with 0.1% BSA and 0.3% Triton X-100 and were rinsed in 0.1% BSA, 0.2% gelatine, and 0.05% saponin.

For bright field microscopy, sections were incubated for 1 h with horseradish peroxidase conjugated secondary antibody in 0.1% BSA, 0.3% Triton X 100 in PBS and were washed in 0.1% BSA, 0.2% gelatine, and 0.05% saponin in PBS before visualization with diaminobenzidine in 33% H_2_O_2_ for 10 min. Finally, the sections were counterstained with Meyers hematoxylin and were rinsed in running tap water before dehydration in graded ethanol and xylene and mounting with coverslips using Eukitt (Merck Life Science, Søborg, Denmark). For immunofluorescence staining, the blocking of peroxidase was omitted, and fluorophore-tagged secondary antibodies (Alexa 488 and Alexa 555) were applied (Fisher Scientific, Slangerup, Denmark). Nuclei were stained with Topro3 (Invitrogen, Taastrup, Denmark). Coverslips were mounted with a hydrophilic mounting medium containing glycergel antifading reagent (DAKO, Glostrup, Denmark).

### 4.3. Cell Polarity Markers

An array of primary antibodies was applied to assess the basic cell polarity (Table 2). The proteins Syntaxin-3, Crumbs3a, Pals-1, PATJ, Par-6, and CDC-42 were used as apical membrane domain markers, while E-cadherin, Scribble, and Lgl-1 were used as basolateral membrane domain markers.

### 4.4. Microscopy and Image Processing

Bright field microscopy was performed on a Leica DM2500 equipped with a Leica MC170 HD camera, PL Fluotar 25×/0.75 N/A and PL Apo 63×/1.32 NA oil immersion objectives. Fluorescence imaging was performed using a DM IRE2 inverted laser scanning confocal microscope (Leica Microsystems, Wetzlar, Germany) equipped with an HCX PC APO CS 63×/1.32 NA oil immersion objective or an inverted 710 laser scanning confocal microscope (Zeiss, Oberkochen, Germany) equipped with a Plan-Apochromat 63×/1.4 NA oil immersion objective. Images were acquired with an 8-bit image depth, 1024 × 1024 pixel resolution, with an image averaging of 3–4 frames and processed in Image-J software (version 1.52a, National Institutes of Health, MD, USA, http://imagej.nih.gov/ij).

Initial microscopy of stained sections was carried out by one investigator, who located cysts of interest to then be checked and validated with regards to polarity by a second investigator. A cyst was deemed a cyst only when the lumen was dilated significantly beyond what would be expected from normal tubules. Both biopsies were analyzed by bright field microscopy, while only one of these was used for immunofluorescence microscopy. In each section, all cysts were inspected, and planar epithelial cysts were excluded from analysis, as they did not display immunoreactivity to our antibodies. 

## Figures and Tables

**Figure 1 ijms-25-01904-f001:**
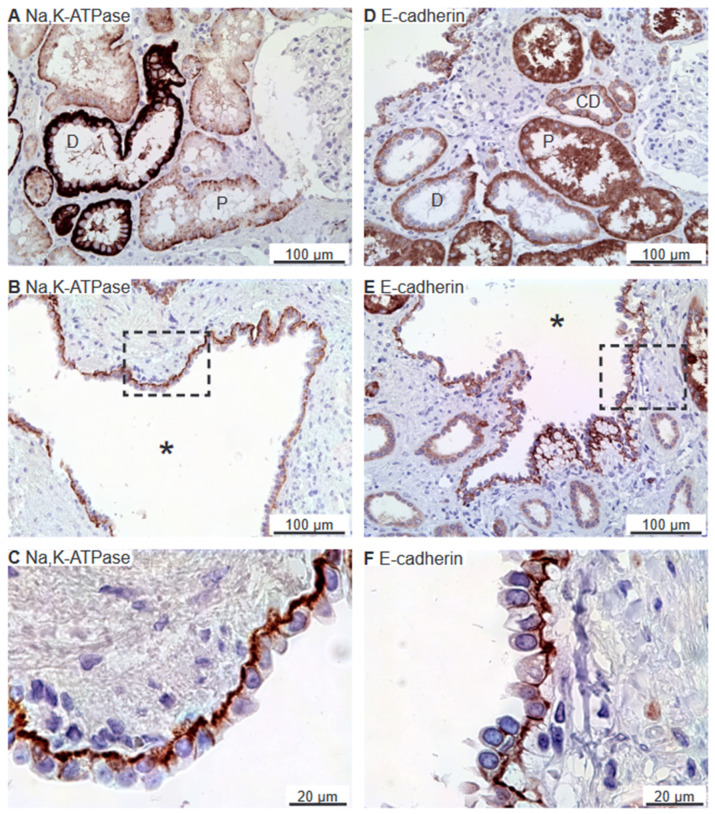
Immunohistochemical analysis of renal Na,K-ATPase α1 and E-cadherin expression in normal human kidney and hPKD tissue. (**A**) Immunoperoxidase overview image of normal tubules. (**B**) Image of hPKD kidney tissue showing a cyst (asterisk). (**C**) Magnification of the window marked in panel (**B**) showing the basolateral distribution of Na,K-ATPase α1 in the cyst epithelium. (**D**) Staining of normal tubules showing basolateral localization of E-cadherin. (**E**) Image of hPKD kidney tissue showing a cyst (asterisk). (**F**) Magnification of the window marked in panel (**E**). E-cadherin is also basolaterally localized in the cyst-lining epithelium. “P” indicates proximal tubules, “D” marks distal tubules, and “CD” points out collecing ducts.

**Figure 2 ijms-25-01904-f002:**
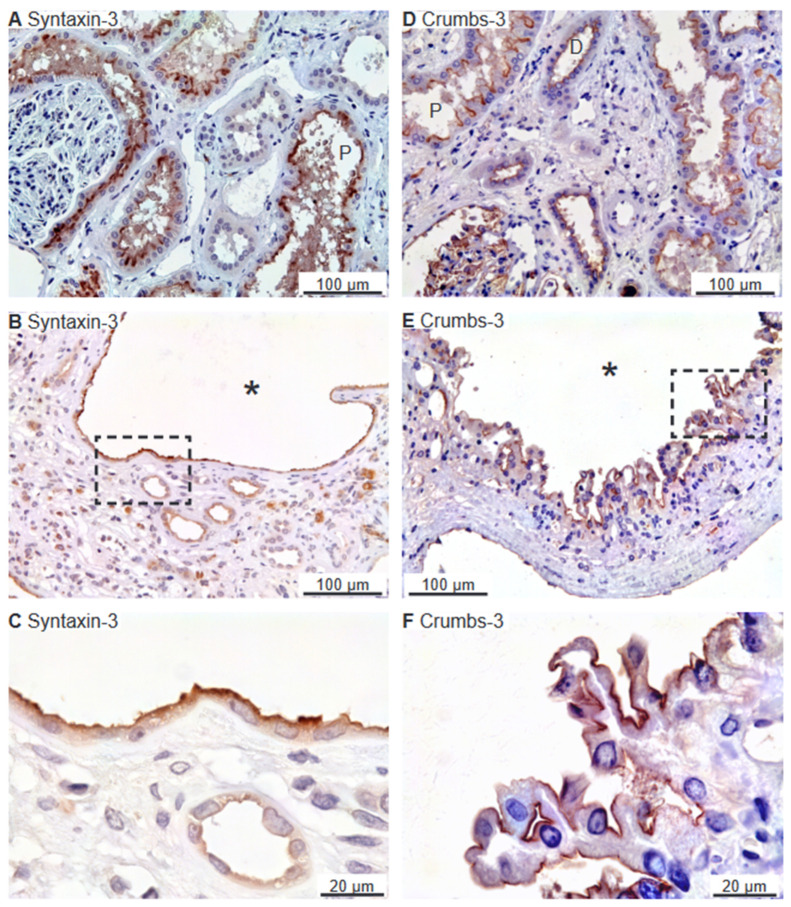
Immunolocalization of Syntaxin-3 and Crumbs-3 in normal human kidney and hPKD tissue. (**A**) Immunolocalization of Syntaxin-3 in normal tubules. (**B**) Image of hPKD kidney tissue showing a cyst (asterisk). (**C**) Magnification of the window in panel (**B**) showing apical localization of Syntaxin-3 in the cyst-lining epithelium. (**D**) Immunolocalization of Crumbs-3 in normal tubules. (**E**) Image of hPKD tissue showing a cyst (asterisk). (**F**) Magnification of the window in panel (**E**) showing apical localization of Crumbs-3 in the cyst epithelium. “P” indicates proximal tubules and “D” marks distal tubules.

**Figure 3 ijms-25-01904-f003:**
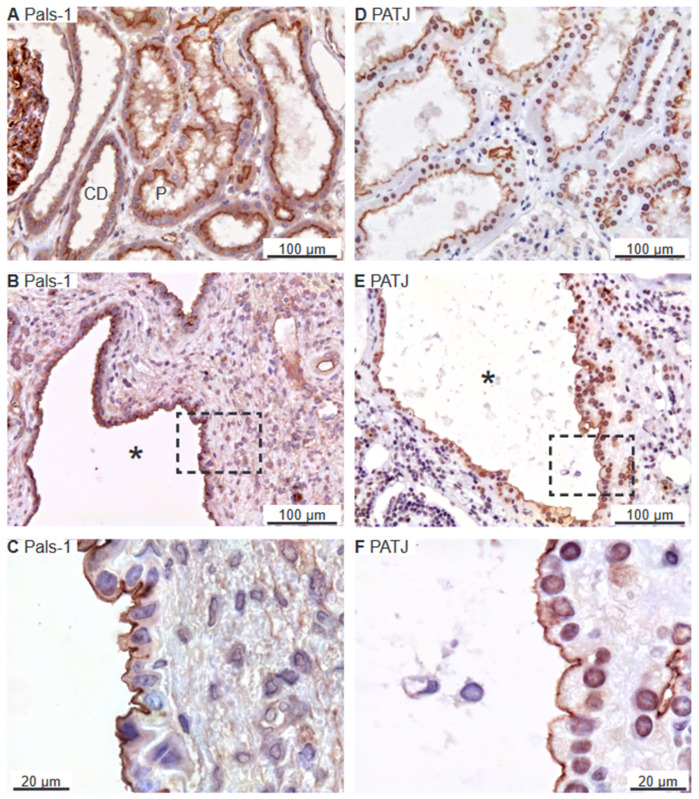
Immunolocalization of Crumbs-complex components Pals-1 and PATJ. (**A**) Staining of normal human tubules showing apical Pals-1 immunoreactivity. (**B**) Image of hPKD tissue stained for Pals-1 showing a cyst (asterisk). (**C**) Magnification of window marked in panel (**B**) showing apical localization of Pals-1 in the cyst epithelium. (**D**) Image of normal human tubules immunostained for apical PATJ. (**E**) Image of hPKD tissue stained for PATJ exemplifying a cyst (asterisk). (**F**) Magnification of the window marked in panel (**E**) showing the apical localization of PATJ in the cyst-lining epithelium. “P” indicates proximal tubules and “CD” points out collecing ducts.

**Figure 4 ijms-25-01904-f004:**
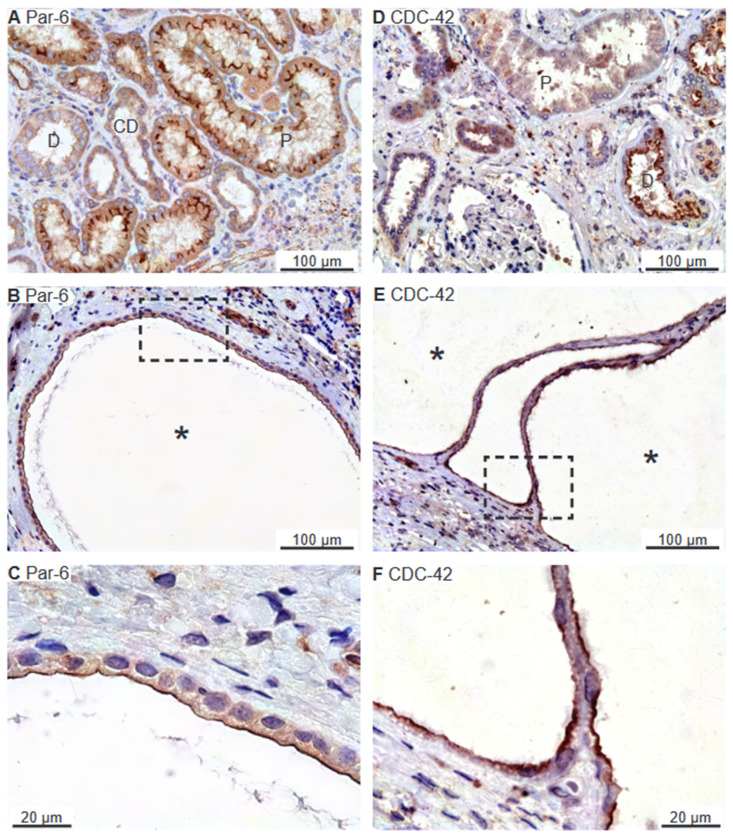
Immunolocalization of renal PAR complex components Par-6 and CDC-42. (**A**) Normal human kidney tubules immunostained for apical Par-6. (**B**) Image of hPKD kidney stained for Par-6 showing a cyst (asterisk). (**C**) Magnification of the window marked in panel (**B**) showing apical distribution of Par-6 in the cyst. (**D**) Normal human kidney tubules stained for apical CDC-42. (**E**) Image of hPKD kidney stained for CDC-42 showing two adjacent cysts (asterisks). (**F**) Magnification of the window marked in panel (**E**) displaying apical immunolocalization of CDC-42 in the cyst epithelium. “P” indicates proximal tubules, “D” marks distal tubules, and “CD” points out collecing ducts.

**Figure 5 ijms-25-01904-f005:**
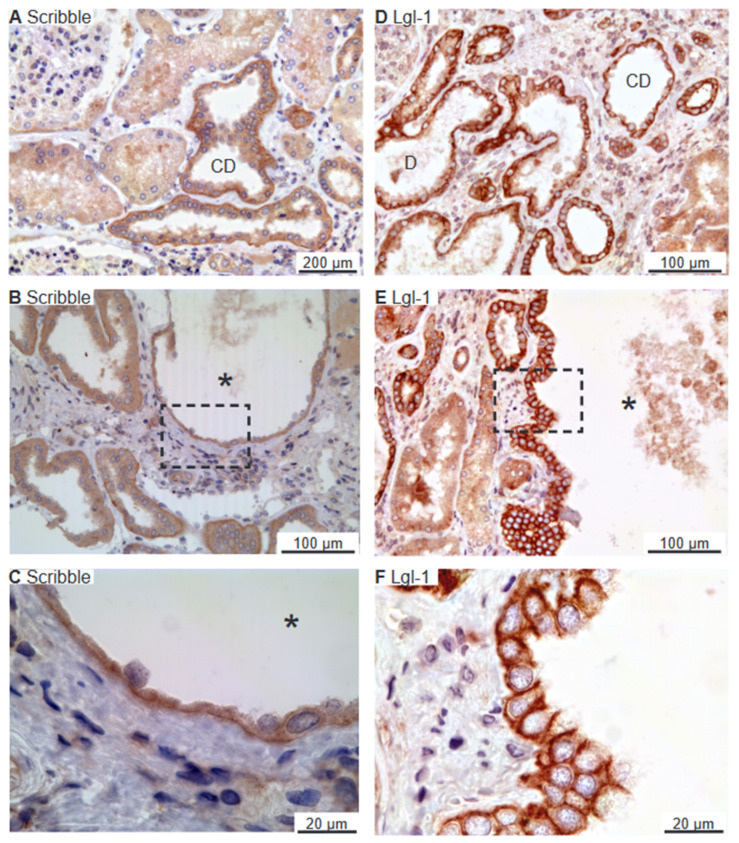
Immunolocalization of renal Scribble complex components Scribble and Lgl-1. (**A**) Overview image of normal human kidney tubules immunostained for Scribble. (**B**) Image of hPKD kidney stained for Scribble showing a cyst (asterisk). (**C**) Higher magnification showing basolateral staining from panel (**B**). (**D**) Image of normal human kidney tubules stained for basolateral Lgl-1. (**E**) Image of hPKD kidney stained for Lgl-1 displaying a cyst (asterisk). (**F**) Higher magnification of the window marked in panel (**E**) showing basolateral localization of Lgl-1 in the cyst-lining epithelium. “D” marks distal tubules and “CD” points out collecing ducts.

**Figure 6 ijms-25-01904-f006:**
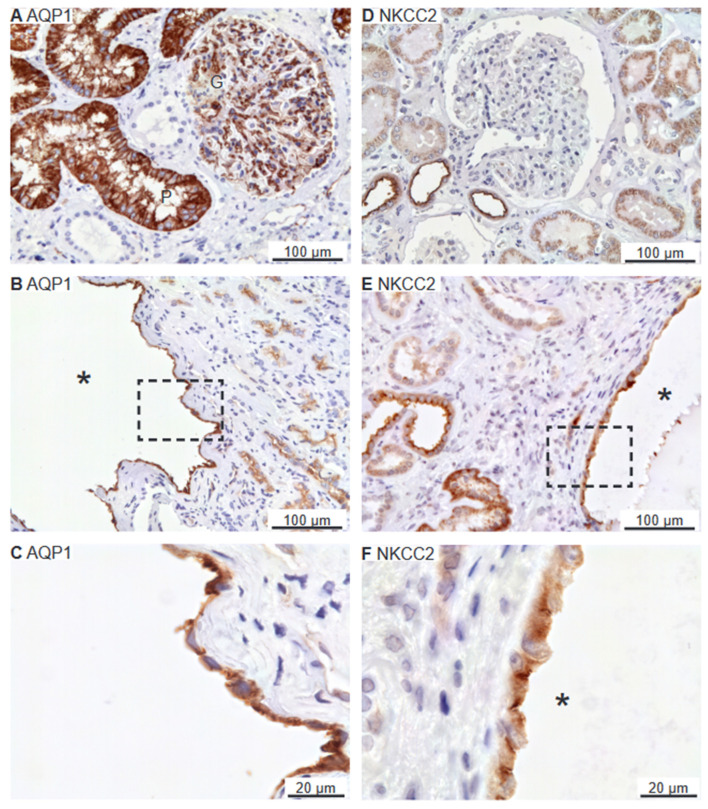
Renal staining for the proximal tubular water channel AQP1 and the thick ascending limb co-transporter NKCC2. (**A**) Image of normal human kidney tissue immunostained for AQP1. (**B**) Overview images of hPKD kidney stained for AQP1 showing a cyst (asterisk). (**C**) Magnification of the window in panel (**B**) showing both apical and basolateral localization of AQP1 in the cyst-lining epithelium. (**D**) Image of normal human kidney tissue immunostained for apical NKCC2. (**E**) Image of renal hPKD tissue stained for NKCC2 showing a cyst (asterisk). (**F**) Magnification of the window in panel (**E**) showing apical staining for NKCC2 in the cyst-lining epithelium. “G” marks glomeruli and “P” indicates proximal tubules.

**Figure 7 ijms-25-01904-f007:**
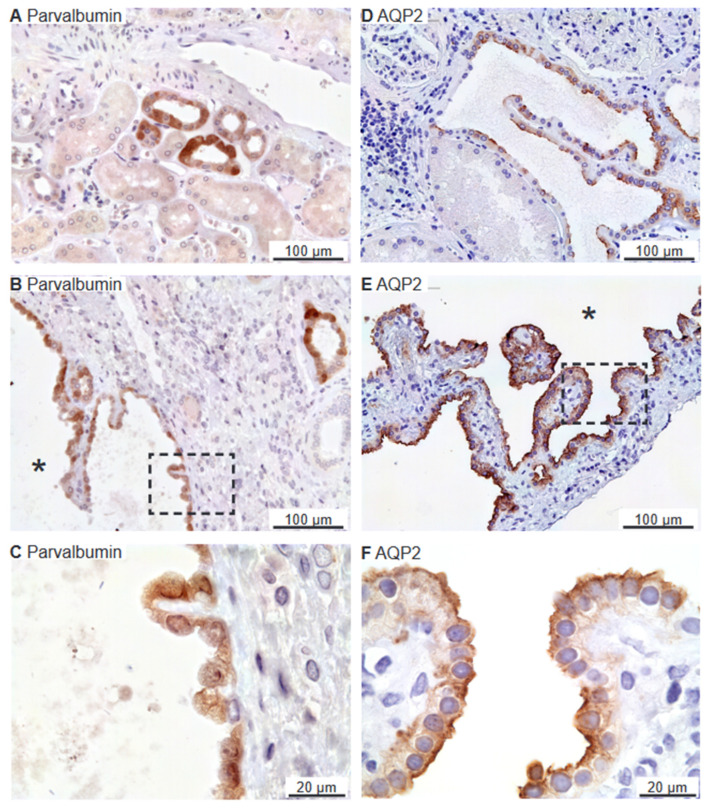
Renal staining for distal convoluted tubule parvalbumin and the collecting duct water channel AQP2. (**A**) Image of normal human kidney immunostained for cytosolic parvalbumin. (**B**) Image of hPKD kidney stained for parvalbumin showing a cyst (asterisk). (**C**) Magnification of the window marked in panel (**B**) showing the cytosolic localization of parvalbumin in the cyst-lining epithelium. (**D**) Image of normal human kidney stained for mainly apical AQP2. (**E**) Image of hPKD tissue stained for AQP2 showing a cyst (asterisk). (**F**) Magnification of the window marked in panel (**E**) showing apical localization of AQP2 in the cyst-lining epithelium.

**Figure 8 ijms-25-01904-f008:**
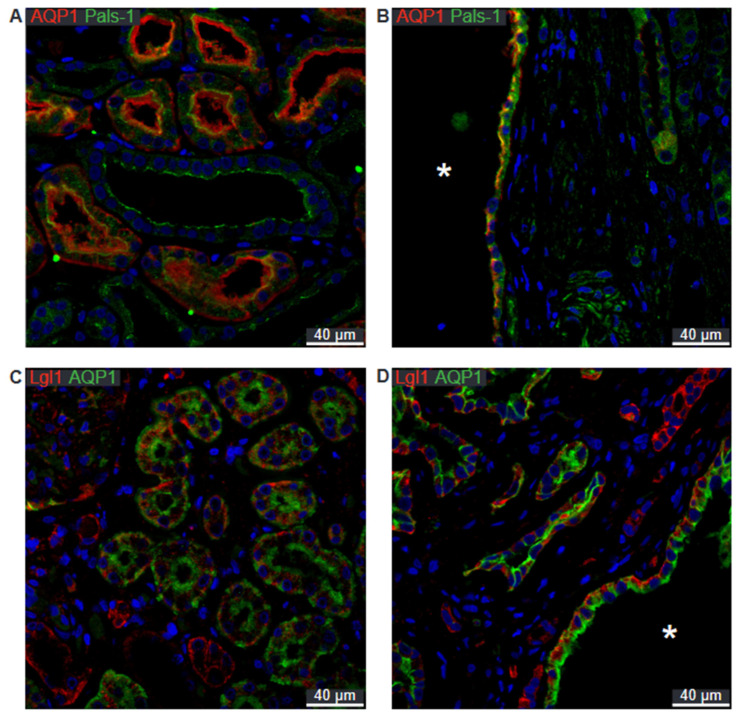
Double immunofluorescence labelling for proximal tubular AQP1 and cell polarity proteins. (**A**) Labelling of AQP1 (red) and Pals-1 (green) in normal human kidney tissue with tubular and cellular colocalization. (**B**) Labelling of AQP1 (red) and Pals-1 (green) in hPKD tissue showing a cyst (asterisk) with similar cellular colocalization. (**C**) Labelling of Lgl-1 (red) and AQP1 (green) in normal human kidney tissue with tubular and cellular colocalization. (**D**) Labelling of Lgl-1 (red) and AQP1 (green) in hPKD tissue showing a cyst (asterisk) with similar cellular colocalization. Nuclei are marked in blue.

**Figure 9 ijms-25-01904-f009:**
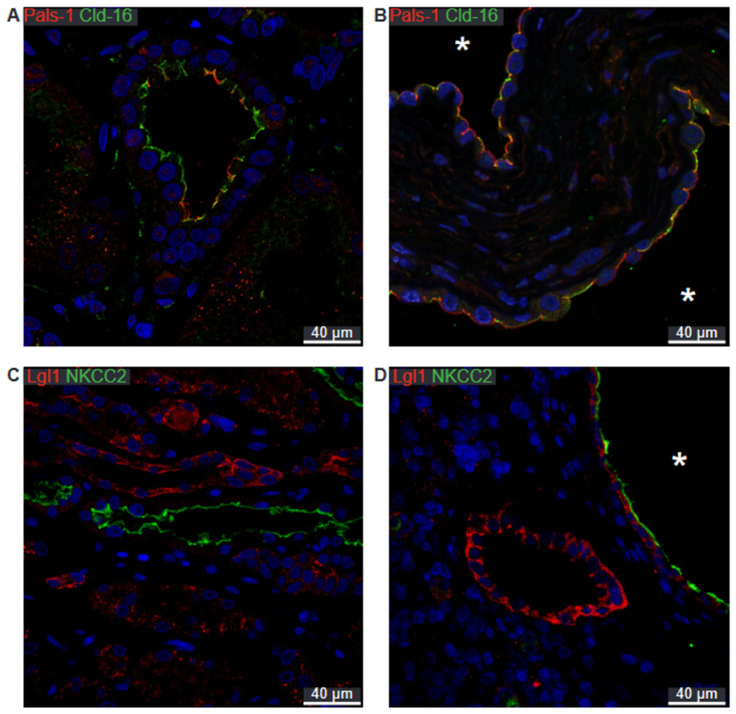
Double immunofluorescence labelling for thick ascending limb NKCC2, Claudin-16, and cell polarity proteins. (**A**) Labelling for Pals-1 (red) and Claudin-16 (green) in normal human kidney tubules. (**B**) Labelling for Pals-1 (red) and Claudin-16 (green) in hPKD kidney showing a cyst (asterisk) with apical localization of both proteins consistent with normal tubular localization. (**C**) Labelling of Lgl-1 (red) and NKCC2 (green) in normal human kidney tissue. (**D**) Labelling of Lgl-1 (red) and NKCC2 (green) in hPKD kidney showing a cyst (asterisk) with apical staining for NKCC2 and basolateral staining for Lgl-1. Nuclei are marked in blue.

**Figure 10 ijms-25-01904-f010:**
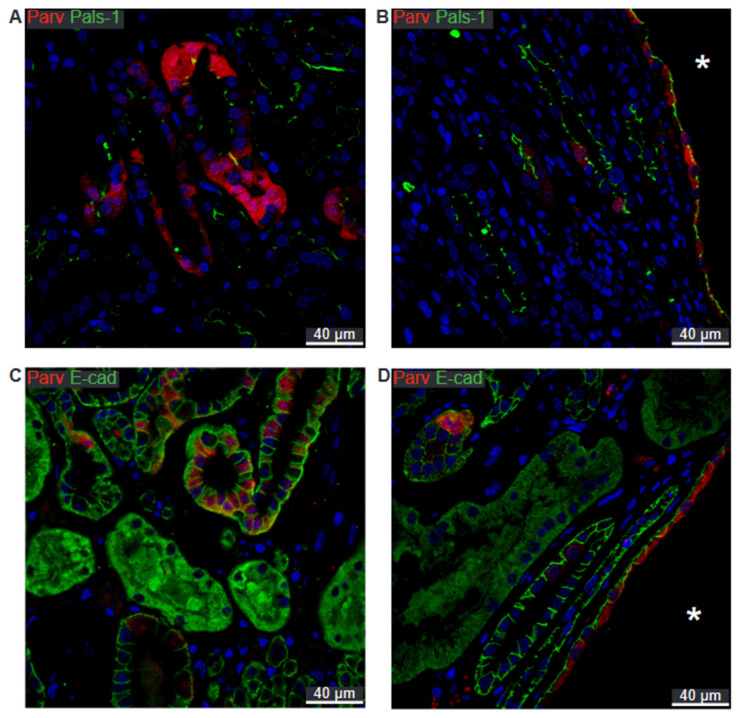
Double immunofluorescence labelling for distal convoluted tubule Parvalbumin and cell polarity proteins. (**A**) Labelling for parvalbumin (Parv, red) and Pals-1 (green) in normal human kidney tissue with tubular and cellular colocalization. (**B**) Labelling for parvalbumin (red) and Pals-1 (green) in hPKD kidney showing a cyst (asterisk) with similar cellular colocalization. (**C**) Labelling for parvalbumin (red) and E-cadherin (E-cad, green) in normal human kidney with tubular and cellular colocalization. (**D**) Labelling for parvalbumin (red) and E-cadherin (green) in hPKD kidney displaying a cyst (asterisk) with similar cellular colocalization. Nuclei are marked in blue.

**Figure 11 ijms-25-01904-f011:**
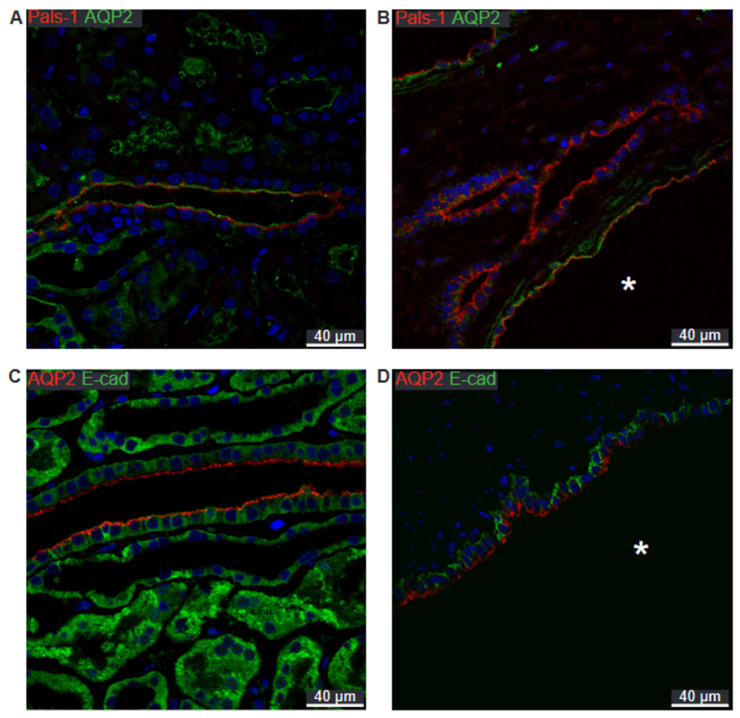
Double immunofluorescence for collecting duct AQP2 and cell polarity proteins. (**A**) Labelling for Pals-1 (red) and AQP2 (green) in normal human kidney with tubular and cellular colocalization. (**B**) Labelling for Pals-1 (red) and AQP2 (green) in hPKD kidney showing a cyst (asterisk) with similar cellular colocalization. (**C**) Labelling for AQP2 (red) and E-cadherin (green) in normal human kidney with tubular and cellular colocalization. (**D**) Labelling for APQ2 (red) and E-cadherin (green) in hPKD kidney showing a cyst (asterisk) with similar cellular colocalization. Nuclei are marked in blue.

**Table 1 ijms-25-01904-t001:** Reactivity of tubule segments by immunoperoxidase. PT: proximal tubule; TAL: thick ascending limb of Henle; DCT: distal convoluted tubule; CCD: cortical collecting duct. “x” indicates the detection of the given cell polarity protein in a specific tubule segment.

Segment	Syntaxin-3	E-Cadherin	Crumbs-3	Pals-1	PATJ	Par-6	CDC-42	Scribble	Lgl-1
PT	x		x	x	x	x			
TAL		x	x	x	x				
DCT		x	x	x	x	x	x	x	x
CCD		x	x	x	x	x		x	x

**Table 2 ijms-25-01904-t002:** Antibodies used in the study.

Target	Antibody No.	Dilution	Host	Source
Na/K-ATPase	3B-0/56-0	1:5000	Mouse	Forbush B III, New Haven, CT, USA
E-cadherin	LS-B12414-300	1:1000	Goat	LS Bio, Seattle, WA, USA
Syntaxin-3	sc-47437	1:25	Goat	Santa Cruz Biotech, Dallas, TX, USA
Crumbs3a	crumbs 3a	1:200	Rat	Massey-Harroche & Le Bivic, Marseilles, France
Pals-1	17710-AP	1:200	Rabbit	Proteintech, Manchester, UK
PATJ	Anti-Patj	1:125	Rabbit	Massey-Harroche & Le Bivic, Marseilles, France
Par-6	sc-166405	1:100	Mouse	Santa Cruz Biotech, Dallas, TX, USA
CDC-42	sc-34314	1:50	Goat	Santa Cruz Biotech, Dallas, TX, USA
Scribble	sc-11048	1:10	Goat	Santa Cruz Biotech, Dallas, TX, USA
Lgl-1	UNC 17-35	1:25	Mouse	Patrick Humbert, Melbourne, Victoria, Australia
AQP1	NB6000-749	1:2000	Rabbit	Novus Biologicals, Abingdon, UK
NKCC2	1495ap	1:200	Rabbit	[44,45]
NCC	SPC-402D	1:10,000	Rabbit	StressMarq Biosciences Victoria, BC, Canada
Parvalbumin	PV235	1:400	Mouse	Swant AG, Burgdorf Switzerland
AQP2	H7661	1:2000	Rabbit	[46]
AQP2	sc-9882	1:250	Goat	Santa Cruz Biotech, Dallas, TX, USA

## Data Availability

Data is contained within the article.

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
