# Peer review of "The Cyst Epithelium in Polycystic Kidney Disease Patients Displays Normal Apical-Basolateral Cell Polarity"

_ijms, 2024, doi:10.3390/ijms25031904_

Round 1

Reviewer 1 Report

Comments and Suggestions for Authors

The manuscript by Sandegaard and colleagues is an elegant and nicely performed study addressing a controversial topic of altered apical-to-basolateral cell polarity in cystic cells as a potential mechanism to drive PKD progression.  For this, the authors employed a systemic comparison of subcellular localization of the major transporting and cell-polarity proteins in untransformed renal tubular and cystic epithelia with immunohistochemistry from 6 different human kidney biopsy samples.  It was demonstrated that there are no detectable differences in normal versus cystic epithelia with respect to the apical/basolateral localization of all tested membrane proteins regardless of the cyst origin (i.e. PT, TAL, DCT, and DC).  The provided images are of satisfactory quality and reflect the major conclusions.  From the negative, it is not specified whether the samples represent both PKD1 and PKD2 mutations; there is no quantitative description of the performed analysis, and cyst sizes, as detailed below:

1.      While the identity of the patients should be protected, it is imperative to specify whether the samples were obtained from patients bearing PKD1 or PKD2 mutations.  It is known that PKD1 mutations cause a more rapid cystogenesis, which does not exclude a possibility of a distorted or even impaired cystic cell polarity in this case.  In contrast, normal cell polarity could be preserved in patients bearing PKD2 mutations, which usually have milder phenotype.  

2.     The presented immunohistochemical images are very convincing.  At the same time, it is necessary to specify the exact numbers of tested cysts/sections/kidneys for each case in the text and respective figure legends to avoid any investigator related bias.

3.     The methods state that the sections were obtained from relatively small portions of the kidneys.  Thus, it is likely that only small millimeter-size (or less) cysts were used for analysis.  Are there any size-dependent differences in small cysts (which are likely yet to be detached) versus occasional large (likely detached) ones?

Author Response

Reviewer 1

From the negative, it is not specified whether the samples represent both PKD1 and PKD2 mutations; there is no quantitative description of the performed analysis, and cyst sizes, as detailed below:

  1. While the identity of the patients should be protected, it is imperative to specify whether the samples were obtained from patients bearing PKD1 or PKD2 mutations. It is known that PKD1 mutations cause a more rapid cystogenesis, which does not exclude a possibility of a distorted or even impaired cystic cell polarity in this case.  In contrast, normal cell polarity could be preserved in patients bearing PKD2 mutations, which usually have milder phenotype. 

Re: We understand the criticism and share the basic curiosity regarding the specific mutations. As it has no consequence for the treatment in Denmark, we do not routinely determine which disease-causing mutations the polycystic patients harbor before surgery. Statistically, the majority (74%) of patients here have PKD1 mutations (PMID 33639313). We have nevertheless explored our possibilities of addressing the point. There are, however, three reasons we unfortunately cannot accommodate the reviewer: 1) Knowing each patient’s specific mutations would have bearings for the course of the disease and the prognosis. This conflicts with the legal terms for using the biopsies and would require a full license for establishing a biobank. 2) Extraction of high quality DNA from the remaining biopsies is uncertain and a following full genetic analysis requires involvement of clinical geneticists with the option of full genomic sequencing and massive analysis to identify relevant disease-causing mutations (approximately 13% of patients do not have PKD1 and PKD2 mutations in the referred study). 3) The price of the described analysis would be 2,000 US$ per biopsy, which is quite high in relation to the focus of the study. The study relates to previous literature on cyst epithelial polarity in polycystic kidney disease, where no information on specific mutations or genetic data were systematically presented. Thus, our findings or conclusions would not benefit from the information.

  1. The presented immunohistochemical images are very convincing. At the same time, it is necessary to specify the exact numbers of tested cysts/sections/kidneys for each case in the text and respective figure legends to avoid any investigator related bias.

Re: We acknowledge the concern and have discussed the same issues along the course of the study. As the biopsies represent only a small fraction of the kidneys, we did not find it fruitful to perform statistical analysis. The clear tendency was that the larger cysts had planar epithelium lining, while the smaller had cuboidal epithelium. We have clarified and discussed the issues in the revised manuscript.

  1. The methods state that the sections were obtained from relatively small portions of the kidneys. Thus, it is likely that only small millimeter-size (or less) cysts were used for analysis.  Are there any size-dependent differences in small cysts (which are likely yet to be detached) versus occasional large (likely detached) ones?

Re: We fully agree that only the relatively small cysts are represented in our analysis. Again, the reason was the lack of immunoreactivity in larger cysts (probably the mature, detached non-secreting cysts) and, thus, the lack of basic cell polarization and membrane transporter expression. This is clarified in the revised manuscript.

Reviewer 2 Report

Comments and Suggestions for Authors

Overall, the manuscript provides a comprehensive investigation into the apical-basolateral cell polarity of cyst-lining epithelia in polycystic kidney disease (PKD). The study explores the expression and localization of key polarity proteins and membrane transport markers in renal cysts from six patients with PKD. Here are some comments and suggestions for improvement:

1.      The title could be more specific to convey the focus of the study. Consider adding terms like "immunohistochemical analysis" or "cell polarity assessment" for clarity.

2.      The abstract is concise but could benefit from summarizing the main findings more explicitly.

3.      The introduction is well-structured and provides a good background on PKD and its genetic basis. However, consider briefly mentioning the significance of studying cell polarity in the context of PKD progression.

4.      The discussion is well-structured and effectively analyzes the findings. However, emphasizing the clinical relevance and potential implications of the study's results for understanding PKD would enhance the paper.

5.      Discuss any limitations of the study, such as the small sample size, potential biases, or technical constraints in the immunohistochemical analysis.

6.      Provide more details on the selection criteria for the six PKD patients in the methods section, including any relevant clinical information if appropriate. Clarify how the assessment of cyst polarity was conducted and validated.

Author Response

Reviewer 2

  1. The title could be more specific to convey the focus of the study. Consider adding terms like "immunohistochemical analysis" or "cell polarity assessment" for clarity.

Re: We thank for the constructive suggestions, but prefer to keep the title as it is.

  1. The abstract is concise but could benefit from summarizing the main findings more explicitly.

Re: Again, we appreciate the suggestion. We have introduced a summarizing sentence in the abstract.

  1. The introduction is well-structured and provides a good background on PKD and its genetic basis. However, consider briefly mentioning the significance of studying cell polarity in the context of PKD progression.

Re: We fully agree with the point and have included a brief mention of the relevance of cell polarity in PKD progression.

  1. The discussion is well-structured and effectively analyzes the findings. However, emphasizing the clinical relevance and potential implications of the study's results for understanding PKD would enhance the paper.

Re: We have sought to accommodate the reviewer on this point to the best of our abilities.

  1. Discuss any limitations of the study, such as the small sample size, potential biases, or technical constraints in the immunohistochemical analysis.

Re: We are grateful for the suggestion and have expanded the discussion to accommodate the reviewer.

  1. Provide more details on the selection criteria for the six PKD patients in the methods section, including any relevant clinical information if appropriate. Clarify how the assessment of cyst polarity was conducted and validated.

Re: We have given the details on selection in the method section (six consecutive nephrectomies due to PKD), but have no access to clinical information on the patients as it would conflict with the legal terms for using the biopsies. We have introduced a short passage in the methods summarizing the apical vs. basolateral membrane domain markers.